# Effect of Heat Treatment on the Mechanical and Corrosion Properties of Mg–Zn–Ga Biodegradable Mg Alloys

**DOI:** 10.3390/ma14247847

**Published:** 2021-12-18

**Authors:** Viacheslav Bazhenov, Anastasia Lyskovich, Anna Li, Vasily Bautin, Alexander Komissarov, Andrey Koltygin, Andrey Bazlov, Alexey Tokar, Denis Ten, Aigul Mukhametshina

**Affiliations:** 1Casting Department, National University of Science and Technology “MISiS”, 119049 Moscow, Russia; nastya719ls999@gmail.com (A.L.); misistlp@mail.ru (A.K.); 2Laboratory of Hybrid Nanostructured Materials, National University of Science and Technology “MISiS”, 119049 Moscow, Russia; anna23-95@mail.ru (A.L.); komissarov@misis.ru (A.K.); tokarb2005@mail.ru (A.T.); teden92@yandex.ru (D.T.); much.aigul@gmail.com (A.M.); 3Department of Metallurgy Steel, New Production Technologies and Protection of Metals, National University of Science and Technology “MISiS”, 119049 Moscow, Russia; bautin@list.ru; 4Laboratory of Advanced Green Materials, National University of Science and Technology “MISiS”, 119049 Moscow, Russia; bazlov@misis.ru; 5Laboratory for Mechanics of Advanced Bulk Nanomaterials for Innovative Engineering Applications, St. Petersburg State University, 199034 St. Petersburg, Russia

**Keywords:** biodegradable materials, magnesium alloys, Mg–Zn–Ga, heat treatment, corrosion rate, mechanical properties

## Abstract

Mg alloys have mechanical properties similar to those of human bones, and have been studied extensively because of their potential use in biodegradable medical implants. In this study, the influence of different heat treatment regimens on the microstructure and mechanical and corrosion properties of biodegradable Mg–Zn–Ga alloys was investigated, because Ga is effective in the treatment of disorders associated with accelerated bone loss. Solid–solution heat treatment (SSHT) enhanced the mechanical properties of these alloys, and a low corrosion rate in Hanks’ solution was achieved because of the decrease in the cathodic-phase content after SSHT. Thus, the Mg–4 wt.% Zn–4 wt.% Ga–0.5 wt.% Y alloy after 18 h of SSHT at 350 °C (ultimate tensile strength: 207 MPa; yield strength: 97 MPa; elongation at fracture: 7.5%; corrosion rate: 0.27 mm/year) was recommended for low-loaded orthopedic implants.

## 1. Introduction

Mg alloys are used in bone implantology as load-bearing orthopedic temporary implants that gradually dissolve in the human body as the healing process progresses. Their densities and Young’s moduli are similar to those of the cortical bone, and the use of Mg alloys can prevent the stress-shielding effect observed for Ti alloys [1,2,3]. In addition, Mg alloys are promising candidates for developing biodegradable implant materials, because of their good biocompatibility and sufficiently high strength-to-weight ratio [1,4].

According to a previous report [5], the alloy Mg–4 wt.% Zn–4 wt.% Ga and its variants with low concentrations of Ca, Nd, or Y, after undergoing equal-channel angular pressing (ECAP), exhibit a low corrosion rate of approximately 0.2 mm/year in Hanks’ solution, and show excellent mechanical properties.

Ga is a well-known and highly effective component for the treatment of disorders associated with accelerated bone loss, including osteoporosis [6], hypercalcemia [7,8,9], Paget’s disease [10,11], and multiple myeloma [12]. Additionally, Ga possesses antibacterial properties [13,14,15]. These characteristics underline the potential of Ga as an alloying element for biodegradable Mg alloys.

The atomic radii of Mg, Ga, and Zn are 0.160, 0.135, and 0.137 nm, respectively [16]. Owing to their similar atomic radii, Zn and Ga exhibit approximately the same solid–solution strengthening effect on Mg [17,18]. Consequently, the mechanical properties of the as-casted and heat-treated Mg–Ga and Mg–Zn alloys are poorer than those of the deformed Mg–Ga and Mg–Zn alloys [17,18,19].

The Zn- and Ga-rich intermetallic phases act as cathodes with respect to the α-Mg phase, and promote a high corrosion rate. However, the T4 solid–solution heat treatment (SSHT) of the binary Mg–Zn and Mg–Ga alloys improves their corrosion resistance, because of the partial dissolution of the Zn- and Ga-rich phases [20,21,22].

The above-mentioned advantages of Mg–Zn–Ga alloys render them suitable for bone implant applications in osteosynthesis. Furthermore, the reduction in corrosion rate is greater for the heat-treated alloys compared to the alloys after deformation processing. Thus, in this study we investigated the influence of heat treatment on the microstructures, mechanical properties, and corrosion of Mg–Zn–Ga alloys.

## 2. Materials and Methods

### 2.1. Raw Materials and Melting Procedure

Five Mg–Zn–Ga alloy samples (one sample with the addition of Y), each with different Zn and Ga contents in α-Mg and with different amounts of intermetallic phases, were prepared, and their chemical compositions are presented in Table 1. The chemical compositions of the alloys were determined using energy-dispersive X-ray spectroscopy (EDS) on the metallographic sections, with an analysis area of 1 mm^2^; three areas were analyzed for each specimen. High-purity bulk metals—such as Mg (99.95 wt.% purity), Zn (99.995 wt.% purity), and Ga (99.9999 wt.% purity)—as well as a Mg–20 wt.% Y master alloy, were used as the raw materials for preparing the alloys. The surfaces of the raw materials were ground to prevent oxide-induced contamination of the prepared alloys. The melts were prepared using a resistance furnace with a steel crucible coated with BN. The melting was performed under an atmosphere of Ar + 2 vol.% SF_6_ to prevent the ignition of the melt. The resulting melt was purged with Ar at 730–750 °C for 3 min, and the melt was maintained at this temperature for 10 min before pouring into the mold.

### 2.2. Heat Treatment Response Analysis

The response of the alloys to various heat treatment regimens was investigated using rectangular ingots (20 mm × 150 mm × 270 mm) cast into graphite molds. The ingots were cut into bars (20 mm × 150 mm × 20 mm), and one surface of each bar was ground. Subsequently, SSHT was performed at 300, 350, 400, 450, and 500 °C for 3, 8, 13, 18, 23, 28, 33, 38, 43, and 48 h, followed by quenching in water. The last two SSHT temperatures were applied only to MgZn2Ga2. The SSHT temperatures were selected based on the solidus temperatures of the alloys measured using differential scanning calorimetry (DSC; Setaram Labsys). Subsequently, the quenched samples were aged at 150, 200, and 250 °C for 3, 6, 9, 12, 15, 18, 21, 24, 27, 30, 45, and 60 h. The Brinell hardness, electrical conductivity, and quantity of the intermetallic phase in the alloy microstructure were measured after each heat treatment step.

The Brinell hardness was measured using a universal hardness tester (Innovatest Nemesis 9001); the ball (diameter: 2.5 mm) was held under a load of 613 N for 10 s. The electrical conductivities of the alloys were measured using a contact-free eddy current conductivity meter (VE-27NC “Sigma”, Yekaterinburg, Russia) in the measurement range of 5.0–37.0 MS/m.

### 2.3. Microstructural Observations and Thermal Analysis

The microstructural observations and EDS analysis were performed using a Tescan Vega SBH3 scanning electron microscope (SEM) equipped with an EDS system (Oxford Instruments, Abingdon, UK). The area occupied by phases in the SEM image was calculated using Tescan software to determine the intermetallic-phase volume fraction. The DSC measurements were performed at a heating rate of 20 °C/min in Al_2_O_3_ crucibles under Ar gas flow to measure the liquidus and solidus temperatures of the as-cast alloys.

### 2.4. Mechanical Properties

The samples for the tensile tests were prepared by casting ingots (340 mm × 50 mm × 32 mm) into a graphite mold [23]. Cylindrical samples (diameter: 5 mm; length: 60 mm) were lathe-machined, and tensile tests were performed on both as-casted and heat-treated alloy samples using an 5569 universal testing machine (Instron, Glenview, IL, USA).

### 2.5. Corrosion Testing

Cylindrical ingots (diameter: 30 mm; height: 150 mm) were cast into a steel mold for corrosion testing. SSHT was performed for 18 h at 350 °C (450 °C for MgZn2Ga2) on the alloy samples prior to the corrosion test. The sample disks (each with a diameter of 15 mm, height of 5 mm, and surface area of ~6 cm^2^) prepared by wire-cutting the ingots were used for immersion corrosion tests. Finally, the disks were ground using a 320-grit abrasive SiC paper. The disks were immersed in 400 mL of Hanks’ solution (PanEco, Moscow, Russia) at 37 ± 0.5 °C for 192 h. For each alloy composition, 6–8 samples were tested, and the ratio of the solution volume to the sample surface area was 70 mL/cm^2^. The average corrosion rate (in mm/year) was calculated according to the ASTM standard [24], and 1 mL of the evolved H_2_ gas was converted into 1 mg of weight loss of the samples based on previous reports [19,25,26]. The variation in the pH of the corrosive media was measured using a pH meter (HI83141, Hanna Instruments, Woonsocket, RI, USA).

Disk samples with a diameter of 15 mm and a height of 8 mm were used for the electrochemical corrosion test; the measurements were performed at 37 °C in Hanks’ solution using a IPC Pro MF potentiostat/galvanostat/frequency response analyzer corrosion system (Volta, Saint Petersburg, Russia). The three-electrode system comprised a working electrode made up of alloy samples with an exposure area of 0.4 cm^2^, while Pt and saturated Ag/AgCl electrodes were used as the counter and reference electrodes, respectively. Prior to the electrochemical tests, the alloy samples were immersed in 0.3 wt.% HNO_3_ aqueous solution for 2 s, followed by rinsing with distilled water. Potentiodynamic polarization measurements were performed between the cathodic region at −2.3 V and the anodic region at −1 V, at a scan rate of 1 mV/s; five curves were obtained for each alloy. Tafel fitting was used to measure the corrosion current density and corrosion potential of the alloys, and their corrosion rate was calculated [27].

## 3. Results

### 3.1. Microstructure of as-Cast Alloys

The microstructures of the investigated as-cast alloy samples are shown in Figure 1a–e. The microstructures of Mg−Zn−Ga alloys exhibited dendrites of the Mg solid solution (α-Mg) and a divorced eutectic of α-Mg with intermetallic phases. The amounts of the intermetallic phases in the alloys are shown in Figure 1f. The minimum and maximum intermetallic-phase contents were observed for MgZn2Ga2 (1.3 vol.%) and MgZn6.5Ga2 (5.7 vol.%), respectively. It was expected that the intermetallic-phase content could be increased by increasing the concentrations of the alloying elements.

The backscattered electron images (BSEIs) and EDS maps for MgZn4Ga4 and MgZn4Ga4Y0.5 are presented in Figure 2, which clearly shows that the eutectic contained both Zn- and Ga-rich phases. Quantitative EDS analysis of the intermetallic eutectic phases (Table 2) showed that the Ga-rich phase was Mg_5_Ga_2_, with a small amount of Zn [28,29]. In contrast, the Zn-rich phase could be Mg_7_Zn_3_, and a small amount of Ga was also found in this phase [30]. The addition of Y promoted the formation of a eutectic Y-rich intermetallic phase (Figure 2g) with a Ga/Y atomic ratio close to 1/1 (Table 2), implying that the phase was GaY in accordance with the Ga−Y phase diagram [31]. Thus, the phases found in MgZn4Ga4 were typical of the other investigated Mg−Zn−Ga alloys [5].

### 3.2. DSC Analysis of as-Cast Alloys

The DSC analysis results are shown in Figure 3. The liquidus temperatures were determined from the DSC cooling curves, and the corresponding results are presented in Figure 3a. Notably, the liquidus temperature of the alloys decreased with increasing concentration of the alloying elements. Conversely, the solidus temperature of all of the investigated alloys was 316 °C, which was obtained from the DSC heating curves (Figure 3b). This solidus temperature value was close to that obtained for MgZn4Ga4 (307 °C) in our previous study [5]. As mentioned before, both the Ga- and Zn-rich eutectic phases were observed in the microstructural and EDS analyses of the alloy samples; based on these results, the solidus temperature (316 °C) could be associated with the formation of the L→α-Mg+Mg_7_Zn_3_+Mg_5_Ga_2_ ternary eutectic.

### 3.3. SSHT and Aging Response of Alloys

The DSC analysis revealed the solidus temperature for all of the investigated alloys to be 316 °C. However, the compositions of MgZn4Ga4, MgZn6.5Ga2, and MgZn4Ga2 were close to the maximum solubility of Zn and Ga in α-Mg for the Mg–Zn–Ga system obtained in previous work [5]; moreover, non-equilibrium solidification possibly led to the formation of the eutectic in the microstructure of the alloys. In this case, when the alloy was heated, the non-equilibrium eutectic dissolved, resulting in a higher real solidus temperature of the alloy. Therefore, the SSHT temperatures of 300, 350, and 400 °C were selected for this study. However, the composition of MgZn2Ga2 was far from the solubility limit [5]; hence, no eutectic formed in this alloy under equilibrium conditions. This result was experimentally confirmed by the observed weak-intensity peak of eutectic transition in the DSC curve. SSHT at 450 and 500 °C was also applied to MgZn2Ga2.

The influence of SSHT temperature and time on the intermetallic-phase contents (Mg_7_Zn_3_, Mg_5_Ga_2_, and GaY) and electrical conductivity of the Mg–Zn–Ga alloys are depicted in Figure 4. The decrease in intermetallic-phase quantity was approximately the same at 350 and 400 °C, and the decrease corresponded to the maximum decrease in the intermetallic-phase quantity. After 48 h of SSHT at the aforementioned temperatures, less than 1 vol.% of the intermetallic phases was left in the microstructures of the alloys. The SSHT at 300 °C was less effective, especially for MgZn4Ga4, MgZn4Ga4Y0.5, and MgZn6.5Zn2; approximately 3.5 vol.% of the intermetallic phases remained in the microstructure of these three alloy samples after 48 h of SSHT. Complete intermetallic-phase dissolution and the α-Mg single-phase microstructure were observed in MgZn4Ga2 and MgZn2Ga2 post-SSHT. The hardness of the alloys was also measured; however, the changes in the hardness during SSHT were negligible and, hence, are not discussed in this paper.

The variations in the electrical conductivity during SSHT were in good correlation with the measured intermetallic-phase quantities (Figure 4). The maximum decrease in the electrical conductivity was observed at the SSHT temperatures of 350 and 400 °C. The decrease in the electrical conductivity during SSHT was associated with an increase in the Zn and Ga contents in α-Mg, because these solute elements act as barriers that hinder the free path of electrons and phonons. Based on the electrical conductivity measurements, the maximum solid–solution hardening effect could be observed in MgZn4Ga4 and MgZn4Ga4Y0.5, with the lowest electrical conductivity of approximately 9 MS/m and the maximum solute elements fraction in α-Mg. For MgZn2Ga2, the electrical conductivity increased when SSHT was applied at 350 and 400 °C, possibly due to precipitation; thus, the SSHT temperature must be higher for this alloy. Based on these observations, SSHT for 18 h at 450 °C was applied to MgZn2Ga2, while SSHT for 18 h at 350 °C was applied to the other investigated alloys in this study.

The influence of aging temperature and time on the hardness and electrical conductivity of the Mg–Zn–Ga alloys is shown in Figure 5. Prior to aging, the alloys were subjected to SSHT for 18 h at 350 °C (at 450 °C for MgZn2Ga2), and then quenched in water. Subsequently, the alloy samples were aged at temperatures of 150, 200, and 250 °C. The results indicate that all of the alloys, except for MgZn2Ga2, exhibited changes in their hardness and electrical conductivity during the aging process, and are hence susceptible to heat treatment aging. The hardness and electrical conductivity increased owing to the formation of Zn- and Ga-rich precipitates during the aging. At the aging temperature of 200 °C, the hardness and electrical conductivity increased by 18 HB and 5 MS/m, respectively, compared to those under the SSHT conditions. For most of the alloys, a longer aging time (up to 60 h) was needed at a lower aging temperature (150 °C) in order to reach the hardness that could be achieved during 9 h of aging at 200 °C. Furthermore, aging at 250 °C resulted in a lower hardness and electrical conductivity than those obtained from aging at 200 °C. These differences were probably observed because of the formation of large and less effective precipitates at 250 °C. The peak aging time could not be determined from the graphs presented in this paper; however, at the most promising aging temperature of 200 °C, 9 h of aging was sufficient to ensure near-maximum hardness.

### 3.4. Microstructure and Mechanical Properties of MgZn4Ga4 and MgZn4Ga4Y0.5 after T4 and T6 Heat Treatments

Two T4 heat treatment regimens—i.e., 18 h of SSHT at 300 °C or 350 °C with water quenching—and two T6 heat treatment regimens—i.e., 18 h of SSHT at 300 °C or 350 °C with water quenching and 9 h aging at 200 °C—were applied to MgZn4Ga4 and MgZn4Ga4Y0.5. These two alloy samples were chosen because they were highly alloyed and showed higher heat treatment responses than did the other investigated alloys. The microstructures and amounts of the Mg_7_Zn_3_ and Mg_5_Ga_2_ intermetallic phases in the as-cast MgZn4Ga4 and T4 and T6 heat-treated MgZn4Ga4 are presented in Figure 6. When the SSHT temperature was 300 °C, the amount of intermetallic phases in the alloy decreased slightly. Additionally, this SSHT regimen led to the formation of precipitates at the dendritic cell boundaries, which were rich in Zn and Ga due to dendritic microsegregation (Figure 6b). The SSHT at 350 °C was more effective because the amount of intermetallic phases after heat treatment was less than 1 vol.%, and the intermetallic phases were partially spheroidized. The aging at 200 °C, after SSHT at 300 °C, produced a negligible effect on the alloy microstructure, as evident from the SEM observations (Figure 6d). However, when aging was applied after SSHT at 350 °C, small precipitates were formed over the entire area of α-Mg.

The microstructures and amounts of the eutectic phases in the as-cast MgZn4Ga4Y0.5 and T4 and T6 heat-treated MgZn4Ga4Y0.5 are shown in Figure 7. The influence of heat treatment on the microstructure of MgZn4Ga4Y0.5 was the same as that on the microstructure of MgZn4Ga4. The only difference was the presence of the GaY phase, which was not dissolved during the SSHT.

The engineering stress–strain curves and mechanical properties of the as-cast MgZn4Ga4 and MgZn4Ga4Y0.5, as well as T4 and T6 heat-treated alloys, are shown in Figure 8. For both of the alloys, the same decreasing trend was observed in their yield strength (YS), and the same increasing trend was obtained in the elongation at fracture (El) after the SSHT. However, the SSHT-induced change in the properties was negligible, and the El increased by only 2%. The aging at 200 °C after SSHT—especially when the SSHT temperature was 350 °C—led to an increase in the YS and a decrease in the El because of the formation of Ga- and Zn-rich precipitates observed in the microstructural analysis. Furthermore, the ultimate tensile strength (UTS) of MgZn4Ga4 and MgZn4Ga4Y0.5 showed almost no changes after the heat treatment, and the corresponding values remained close to 200 MPa for both of these alloys under all conditions.

### 3.5. Microstructure and Mechanical Properties of T4 Heat-Treated Mg–Zn–Ga Alloys

For potential applications in bone implants, the alloy should exhibit an E1 close to 10% [32,33] and a low corrosion rate in order to allow a sufficient biodegradation time for the bone-fracture healing. Considering these practical requirements, the T4 heat treatment regimen of 18 h of SSHT at 350 °C (450 °C for MgZn2Ga2) was selected for the investigated alloys. This heat treatment provides maximum El and a minimal amount of intermetallic phases, which can act as cathodes with respect to α-Mg during the corrosion process.

The microstructures and amounts of intermetallic phases in the Mg–Zn–Ga alloys after 18 h of SSHT at 350 °C (450 °C for MgZn2Ga2) and water quenching are presented in Figure 9. The microstructures of MgZn4Ga4 and MgZn4Ga4Y0.5 were analyzed previously (Figure 6 and Figure 7, respectively), and the amounts of intermetallic phases in these alloys were found to be 0.9 and 1.3 vol.%, respectively. For MgZn6.5Ga2 with the maximum Zn content, spheroidization of the intermetallic phases was not observed (Figure 9c). Furthermore, the remaining content of the intermetallic phases in the alloy after T4 heat treatment was the maximum (2.1 vol.%) among those in the other investigated alloys. In contrast, the amount of intermetallic phases in MgZn4Ga2 and MgZn2Ga2 was close to zero after the T4 heat treatment (Figure 9d,e), and a lower corrosion rate could be expected for these alloys.

The engineering stress–strain curves and mechanical properties obtained for the Mg–Zn–Ga alloys after 18 h of SSHT at 350 °C (450 °C for MgZn2Ga2), and subsequent water quenching, are shown in Figure 10. The same mechanical properties were observed for MgZn4Ga4 and MgZn4Ga4Y0.5. This implies that the addition of Y had no effect on the mechanical properties of MgZn4Ga4. MgZn6.5Ga2 exhibited a lower YS, higher El, and a UTS = 242 MPa, which was the maximum among those of the other alloys. Decreasing the Zn content in the alloys (MgZn4Ga2 and MgZn2Ga2) decreased the YS and UTS, and increased the El. As shown in Figure 9, these alloys had α-Mg single-phase microstructures, and the tensile properties of these alloys were fully promoted by the solid–solution hardening effect of Zn and Ga in α-Mg.

### 3.6. Corrosion Properties of T4 Heat-Treated Mg–Zn–Ga Alloys in Hanks’ Solution

The amount of H_2_ released during the 192 h immersion corrosion testing of the Mg–Zn–Ga alloys in Hanks’ solution after 18 h of SSHT at 350 °C (450 °C for MgZn2Ga2) is shown in Figure 11a. Evidently, the amount of released H_2_ varied significantly for the alloy samples; consequently, a large number of samples were used for the corrosion test. The calculated corrosion rate (CR) of the investigated alloys and the pH of corrosive media (Hanks’ solution) measured at the end of the corrosion tests are shown in Figure 11b. These results show that the addition of 0.4 wt.% Y to MgZn4Ga4 decreased its CR from 0.59 to 0.27 mm/year, i.e., it more than halved the initial CR. For the alloys with a Zn/Ga ratio (wt.%) close to 1, the corrosion rate was approximately the same, i.e., ~0.6 mm/year. When the Zn/Ga ratio was increased to 2 and 3.25 while the Zn content in the alloys was higher, their CR increased to 0.78 and 1.03 mm/year, respectively. The pH of the corrosive media for the high-alloyed MgZn4Ga4, MgZn4Ga4Y0.5, and MgZn6.5Ga2 was close to 9 after 192 h of corrosion testing, while that of the low-alloyed MgZn4Ga2 and MgZn2Ga2 was close to 8. In general, a higher CR is proportional to a higher pH value of the medium; however, this trend was not observed in our results [34,35]. At the same time, it can be seen that pH is higher for high-alloyed alloys, and it is possible that the content of Zn and Ga in the medium affects its pH.

The polarization curves of the Mg–Zn–Ga alloy samples obtained in Hanks’ solution after 18 h of SSHT at 350 °C (450 °C for MgZn2Ga2) are shown in Figure 12a. The anodic current densities were similar for all of the investigated alloys; however, the cathodic current densities showed wide variations. The maximum and minimum cathodic currents were observed for MgZn6.5Ga2 and MgZn2Ga2, respectively. The cathodic reaction became kinetically easier as the amount of cathodic Mg_7_Zn_3_ and Mg_5_Ga_2_ intermetallic phases in the alloy increased which, in turn, promoted a higher CR. The corrosion potential (*E_corr_*) and CR of the alloys determined using Tafel fitting are shown in Figure 12b,c, respectively. A more negative *E_corr_* (−1.55 V) was observed for the alloys with a Zn/Ga ratio (in wt.%) close to 1 (MgZn4Ga4 and MgZn2Ga2). Increasing the Zn/Ga ratio to 2 and 3.25 (MgZn4Ga2 and MgZn6.5Ga2, respectively) caused a positive shift in the *E_corr_*, to −1.51 and −1.40 V, respectively. The increase in the intermetallic-phase area, which acted as a cathode with respect to α-Mg, was responsible for the more positive *E_corr_* of MgZn6.5Ga2, with the maximum volume fraction of intermetallic phases. MgZn4Ga2 exhibited a more positive *E_corr_* than those shown by the higher alloyed alloy samples; moreover, MgZn4Ga4 contained a higher amount of intermetallic phases than did MgZn4Ga2. This means that in this case, the ratio of Zn and Ga in α-Mg had a major effect on *E_corr_*, and Zn had a greater effect on the corrosion potential than did Ga. The addition of 0.4 wt.% Y to MgZn4Ga4 caused a positive shift in *E_corr_* because of the formation of a cathodic GaY phase in the alloy structure. For the investigated alloys, the CR calculated using the corrosion current density was proportional to *E_corr_*. In other words, a more positive *E_corr_* leads to a higher CR. Nevertheless, the calculated CR were approximately the same for the investigated alloys (1.8–2.4 mm/year), except for MgZn6.5Ga2 (CR = 11.1 mm/year).

## 4. Discussion

The microstructures of all of the investigated as-cast alloys consisted of α-Mg, Mg_5_Ga_2_, and Mg_7_Zn_3_ phases. The addition of Y led to the formation of a GaY phase. As expected, the ternary eutectic transition L→α-Mg+Mg_7_Zn_3_+Mg_5_Ga_2_ occurred in the Mg–Zn–Ga system at a temperature of 316 °C, and a long freezing range near 300 °C was observed for the investigated alloys. This caused shrinkage porosity, which affects the mechanical properties of the alloys.

The investigation results showed that SSHT was effective for intermetallic-phase dissolution, and an α-Mg single-phase microstructure was observed for MgZn4Ga2 and MgZn2Ga2 after SSHT. The solid–solution hardening effect of Ga in α-Mg was unknown; however, for Zn, a solid–solution hardening of 20 MPa/wt.% was observed [36]. It was established that the enrichment of α-Mg with Zn and Ga, and that the dissolution of intermetallic phases decreased the YS and increased the El in comparison to those obtained for the as-cast samples. This indicates that the Ga hardening effect in α-Mg was very low, possibly because of the low lattice distortion due to the small difference between the atomic radii of Ga and Mg [16]. In contrast, aging after SSHT slightly increased the YS and strongly decreased the El. The overall effect of aging treatment on the mechanical properties of the investigated alloys was inferior because of the low effectiveness of the Zn- and Ga-rich precipitates [28,29,30]. For the Mg–Zn–Ga alloys, a low solubility of Y in α-Mg (~0.05 wt.%) was observed, similar to that observed in a previously reported study [5], and the solid–solution hardening effect of Y in α-Mg was negligible. Consequently, the addition of Y to MgZn4Ga4 did not produce any changes in its heat treatment response or mechanical properties.

A material with high El is required for bone implants [32,33]. The T4 heat treatment of 18 h of SSHT at 350 °C (450 °C for MgZn2Ga2), followed by quenching in water, was optimal for the investigated alloys, because this treatment induced a high El in the investigated alloys. After SSHT, the low-alloyed Mg–Zn–Ga alloys showed higher El values than did the high-alloyed alloys, owing to the absence of brittle intermetallic phases. Moreover, the YS was higher for the high-alloyed alloys than for their low-alloyed counterparts, because of the solid–solution hardening caused by the high Ga and Zn contents in the α-Mg phase. In addition, the intermetallic phases in the high-alloyed MgZn4Ga4 and MgZn4Ga4Y0.5 alloys’ microstructures were spheroidized after the T4 SSHT. This spheroidization also improved the mechanical properties of the alloys. The combination of YS, UTS, and El observed for the MgZn6.5Ga2 alloy may be attributed to the high Zn content and large solid–solution hardening effect [36].

Another requirement of the bone implant materials is a sufficiently low CR to realize the correct biodegradation time for the complete healing of bone fractures. Immersion corrosion test results showed that increasing the Zn content in the alloys decreased their corrosion resistance because of a positive shift in the *E_corr_* caused by an increase in the amount of Mg_7_Zn_3_ intermetallic phase in the alloy structure. This Mg_7_Zn_3_ intermetallic phase acted as a cathode with respect to the α-Mg phase, and promoted micro-galvanic corrosion. In contrast, the Mg_5_Ga_2_ phase probably had a smaller effect on the CR of the alloys; however, further investigations are required in order to ascertain this effect. The corrosion media, after the immersion corrosion tests of the low-alloyed Mg–Zn–Ga alloys, exhibited a low pH value of ~8. A high alkaline pH has a negative effect on the healing process; thus, a more neutral pH of the corrosive medium during the degradation of MgZn4Ga2 and MgZn2Ga2 may improve the healing process [37,38,39].

The electrochemical corrosion test results showed that the high *E_corr_* of the investigated alloys coincided with the high CR calculated using the corrosion current density. With increasing the amounts of intermetallic phases, the cathodic area increased, leading to an increase in the CR due to a micro-galvanic effect. According to the immersion corrosion test results, the lowest CR (0.27 mm/year) was obtained for MgZn4Ga4Y0.5. This result was not consistent with those of the electrochemical corrosion tests, because the addition of Y to MgZn4Ga4 increased the eutectic amount via GaY phase precipitation, which was cathodic with respect to α-Mg. Thus, there is a definite probability that the addition of Y can change the corrosion product density and, in turn, improve the corrosion resistance owing to the shielding effect of the corrosion products.

Table 3 lists the mechanical characteristics (YS, UTS, and El) and calculated CRs of the alloys fabricated in our study, as well as of the other Ga-containing Mg alloys [5,17,18,19,22]. Notably, the mechanical properties and corrosion resistance of the as-cast binary Mg–Ga alloys were poorer than those of the ternary Mg–Zn–Ga alloys after SSHT in this study. This implies that the addition of Zn to the Mg–Ga alloys can enhance the corrosion resistance and mechanical properties of these alloys, but the Zn/Ga ratio (in wt.%) must be not higher than 1 in this case. The mechanical properties of the ternary Mg–Zn–Ga alloys with low additions of Ca, Y, Nd, and MgGa3.5 that were obtained via hot extrusion were better than those of the other alloys investigated in this study. The enhancement in the mechanical properties after hot extrusion occurred due to grain refinement, which produces a significant effect on the properties of Mg alloys because of their high Hall–Petch strengthening coefficient (~300 MPa·μm^1/2^) [33]. Furthermore, MgZn4Ga4Y0.5 exhibited nearly the same CR after T4 heat treatment and hot extrusion. Overall, MgZn4Ga4Y0.5 can be recommended for orthopedic implants, but only in cases where no high loads are present.

## 5. Conclusions

We investigated the influence of heat treatment on the microstructure as well as the mechanical and corrosion properties of Mg–Zn–Ga alloys, in order to assess their feasibility for osteosynthesis applications. The main results can be summarized as follows:
The microstructures of the investigated as-cast Mg–Zn–Ga alloys (MgZn4Ga4, MgZn4Ga4Y0.5, MgZn6.5Ga2, MgZn4Ga2, and MgZn2Ga2) consisted of α-Mg, Mg_5_Ga_2_, and Mg_7_Zn_3_ eutectic phases. MgZn4Ga4Y0.5 also contained a GaY eutectic phase;The Mg–Zn–Ga alloys under investigation exhibited a solidus temperature of 316 °C, which was related to the ternary eutectic L→α-Mg+Mg_7_Zn_3_+Mg_5_Ga_2_. This ternary eutectic transition was responsible for the low SSHT temperatures;Analysis of the alloy heat treatment response showed that the optimal temperature and time for SSHT were 350 °C (450 °C for MgZn2Ga2 alloy) and 18 h, respectively. Furthermore, based on the analysis results, aging at 200 °C for 9 h is recommended for these alloys;The SSHT of the Mg–Zn–Ga alloys induced a high El (up to 15.2 %) and produced a low fraction of intermetallic phases that provided high corrosion resistance. Moreover, aging had a low effect on the alloy strength; however, it also decreased the El value of the alloy;The immersion corrosion test results showed that the alloys with a Zn/Ga ratio (in wt.%) close to 1 exhibited approximately the same CR of ~0.6 mm/year. For the alloys with Zn/Ga ratios of 2 and 3.25, the CR was 0.78 and 1.03 mm/year, respectively. The addition of Y decreased the CR from 0.59 to 0.27 mm/year because of the improvement in the corrosion-product shielding effect.

Thus, MgZn4Ga4Y0.5 after 18 h of SSHT at 350 °C can be recommended for applications in low-loaded bone implants because of its good mechanical properties (UTS = 207 MPa; YS = 97 MPa; El = 7.5%) and low biocorrosion rate (0.27 mm/year).

## Figures and Tables

**Figure 1 materials-14-07847-f001:**
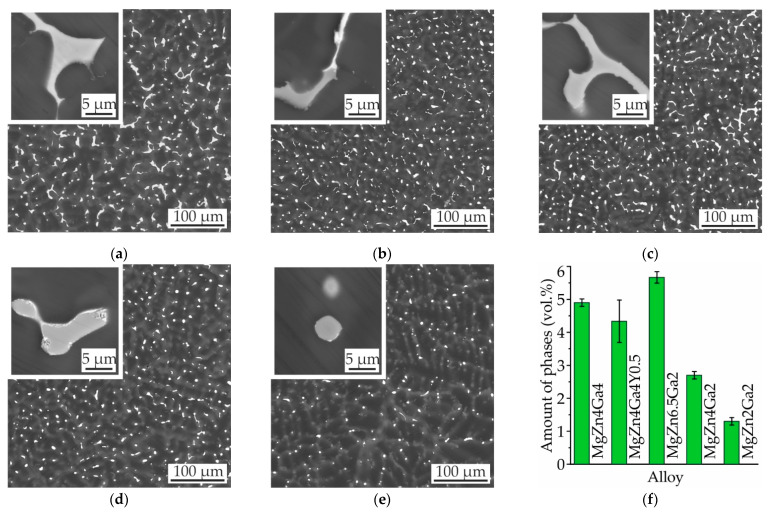
SEM micrographs of (**a**) MgZn4Ga4, (**b**) MgZn4Ga4Y0.5, (**c**) MgZn6.5Ga2, (**d**) MgZn4Ga2, and (**e**) MgZn2Ga2 alloys. (**f**) Intermetallic-phase contents of the as-cast alloys.

**Figure 2 materials-14-07847-f002:**
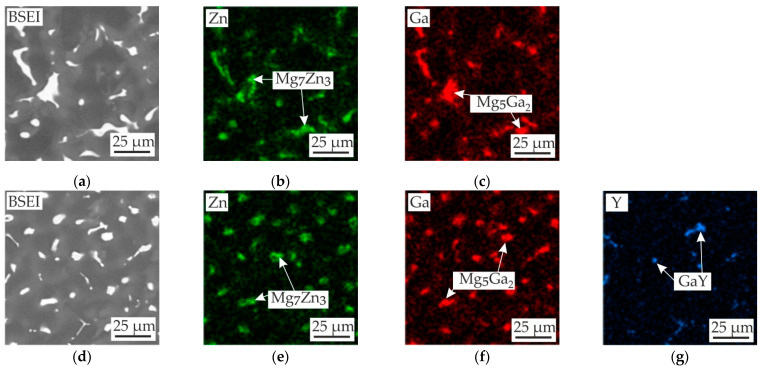
BSEIs (**a**,**d**) and EDS maps showing the distribution of (**b**,**e**) Zn, (**c**,**f**) Ga, and (**g**) Y in the as-cast (**a**–**c**) MgZn4Ga4 and (**d**–**g**) MgZn4Ga4Y0.5.

**Figure 3 materials-14-07847-f003:**
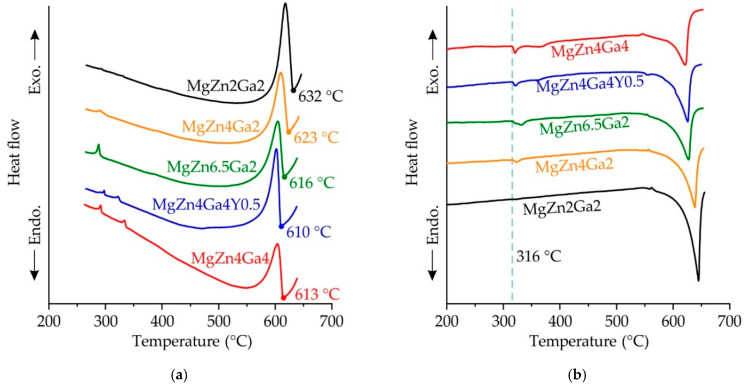
DSC curves of the as-cast Mg−Zn−Ga alloys: (**a**) cooling curves; (**b**) heating curves.

**Figure 4 materials-14-07847-f004:**
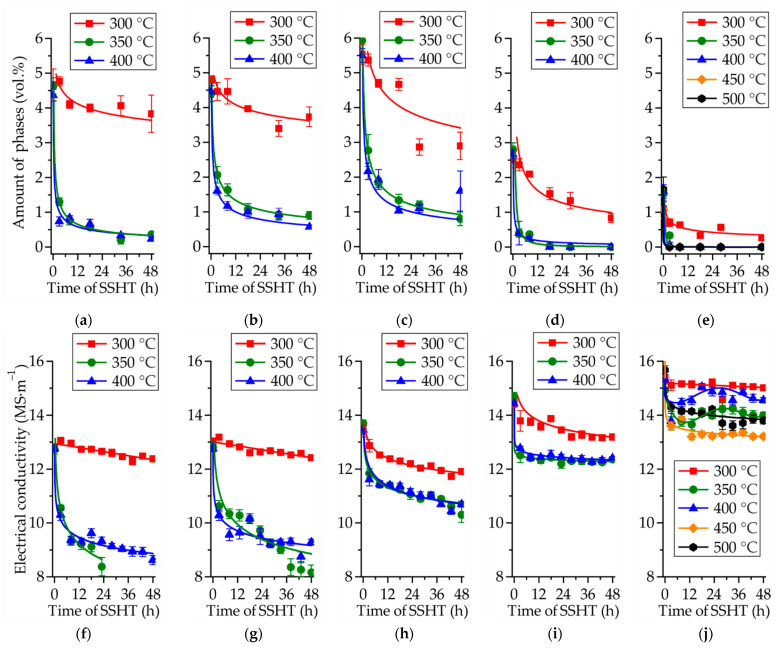
Amounts of intermetallic phases (Mg_7_Zn_3_, Mg_5_Ga_2_, and GaY) (**a**–**e**) and electrical conductivity (**f**–**j**) of the Mg–Zn–Ga alloys: (**a**,**f**) MgZn4Ga4, (**b**,**g**) MgZn4Ga4Y0.5, (**c**,**h**) MgZn6.5Ga2, (**d**,**i**) MgZn4Ga2, and (**e**,**j**) MgZn2Ga2, during SSHT at different temperatures.

**Figure 5 materials-14-07847-f005:**
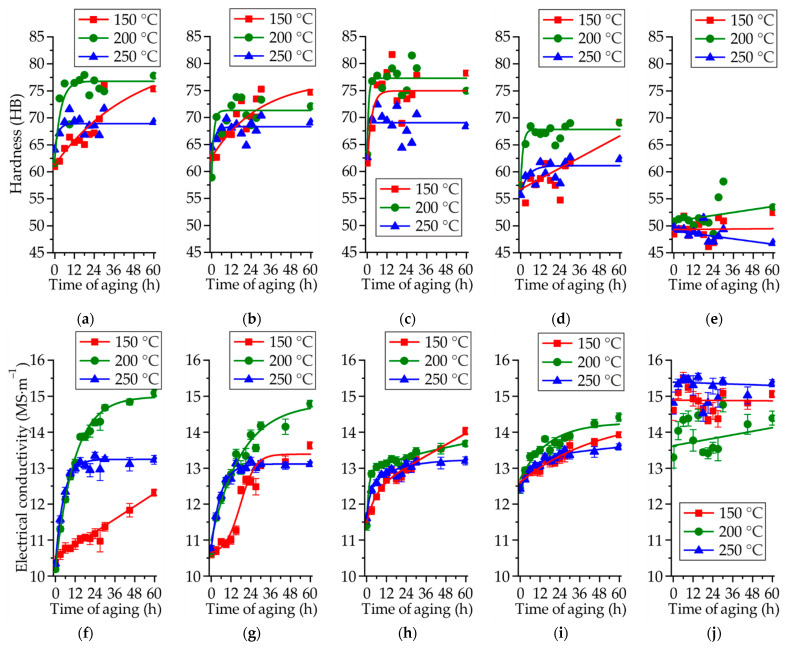
Brinell hardness (**a**–**e**) and electrical conductivity (**f**–**j**) of the Mg–Zn–Ga alloys: (**a**,**f**) MgZn4Ga4, (**b**,**g**) MgZn4Ga4Y0.5, (**c**,**h**) MgZn6.5Ga2, (**d**,**i**) MgZn4Ga2, and (**e**,**j**) MgZn2Ga2 during aging at different temperatures (18 h at 450 °C for MgZn2Ga2, and 18 h at 350 °C for other alloys) after SSHT.

**Figure 6 materials-14-07847-f006:**
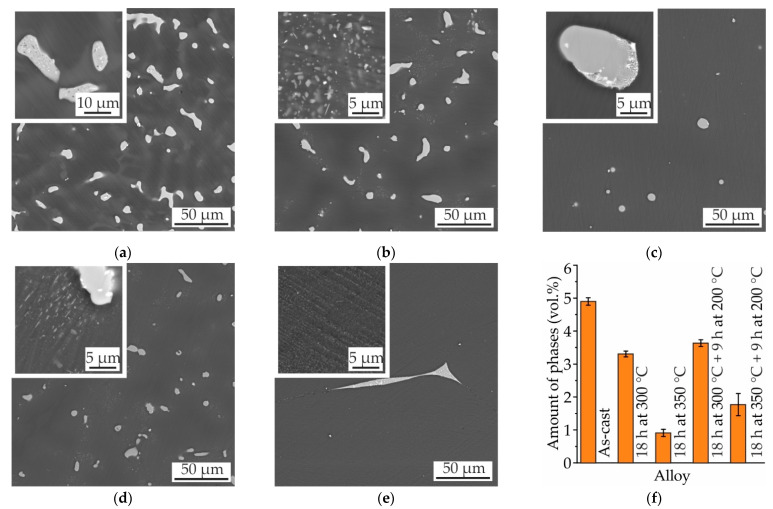
SEM micrographs of MgZn4Ga4: (**a**) as-cast; after T4 heat treatment: (**b**) 18 h at 300 °C, (**c**) 18 h at 350 °C; and after T6 heat treatment: (**d**) 18 h at 300 °C and 9 h at 200 °C, (**e**) 18 h at 350 °C and 9 h at 200 °C; and (**f**) the amount of intermetallic phases (Mg_7_Zn_3_ and Mg_5_Ga_2_) in these conditions.

**Figure 7 materials-14-07847-f007:**
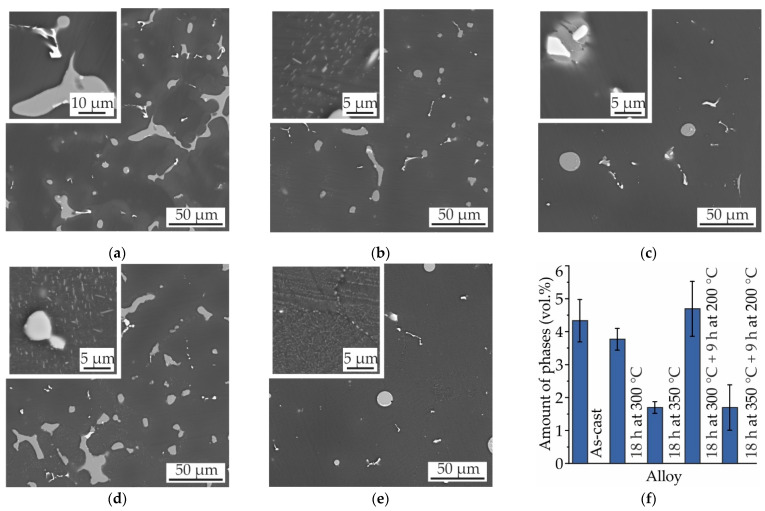
SEM micrographs of MgZn4Ga4Y0.5: (**a**) as-cast; after T4 heat treatment: (**b**) 18 h at 300 °C, (**c**) 18 h at 350 °C; and after T6 heat treatment: (**d**) 18 h at 300 °C and 9 h at 200 °C, (**e**) 18 h at 350 °C and 9 h at 200 °C; and (**f**) the amount of intermetallic phases (Mg_7_Zn_3_, Mg_5_Ga_2_ and GaY) in these conditions.

**Figure 8 materials-14-07847-f008:**
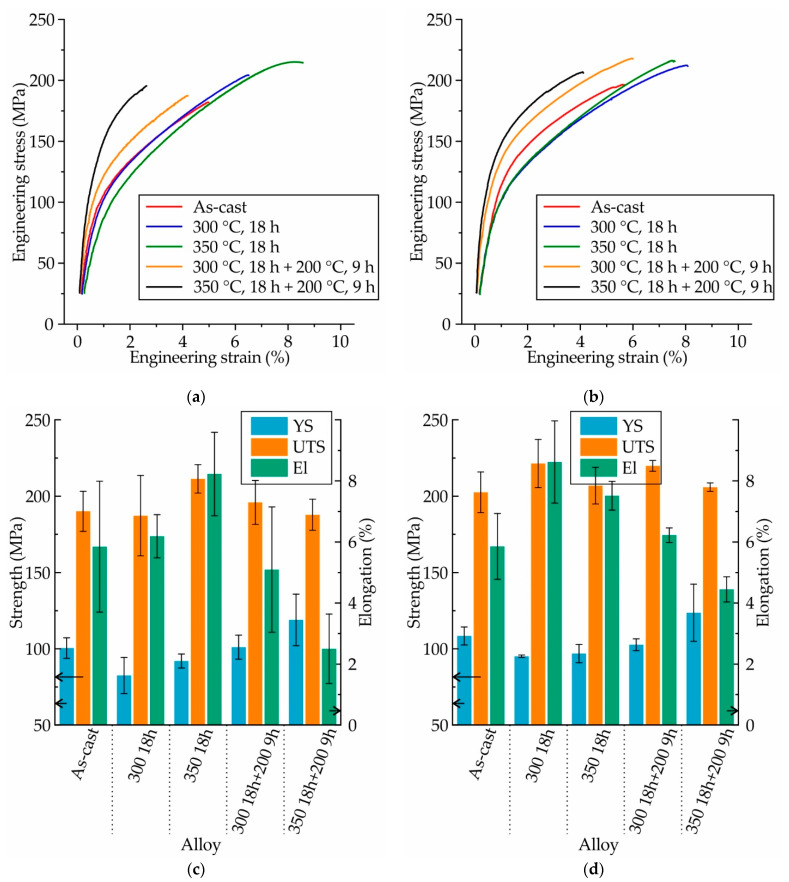
(**a**,**b**) Engineering stress–strain curves and (**c**,**d**) mechanical properties (YS: yield strength; UTS: ultimate tensile strength; El: elongation at fracture) obtained during the tensile testing of the as-cast and heat-treated (**a**,**c**) MgZn4Ga4 and (**b**,**d**) MgZn4Ga4Y0.5.

**Figure 9 materials-14-07847-f009:**
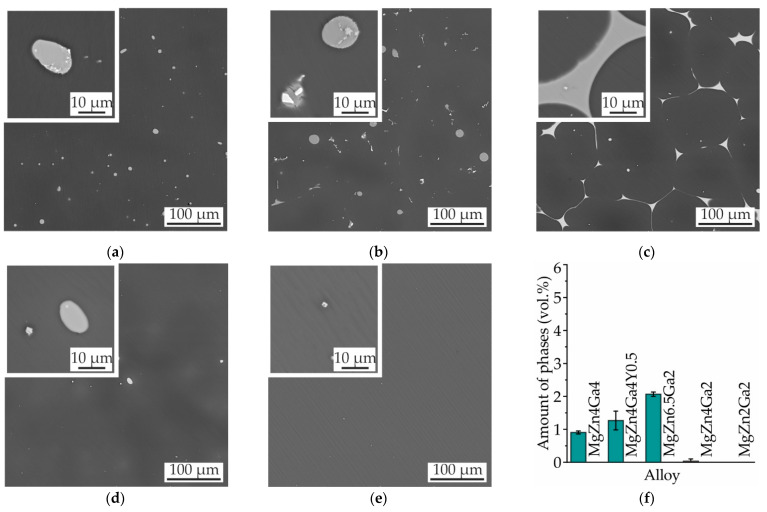
SEM micrographs of (**a**) MgZn4Ga4, (**b**) MgZn4Ga4Y0.5, (**c**) MgZn6.5Ga2, (**d**) MgZn4Ga2, and (**e**) MgZn2Ga2. (**f**) Amounts of intermetallic phases (Mg_7_Zn_3_, Mg_5_Ga_2_, and GaY) in the alloys after T4 heat treatment.

**Figure 10 materials-14-07847-f010:**
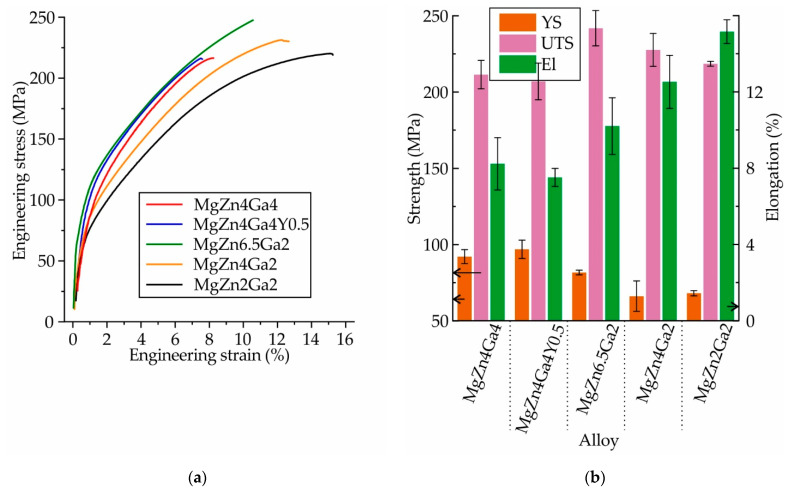
(**a**) Engineering stress–strain curves and (**b**) mechanical properties (YS: yield strength; UTS: ultimate tensile strength; and El: elongation at fracture) obtained during the tensile testing of the T4 heat-treated Mg–Zn–Ga alloys.

**Figure 11 materials-14-07847-f011:**
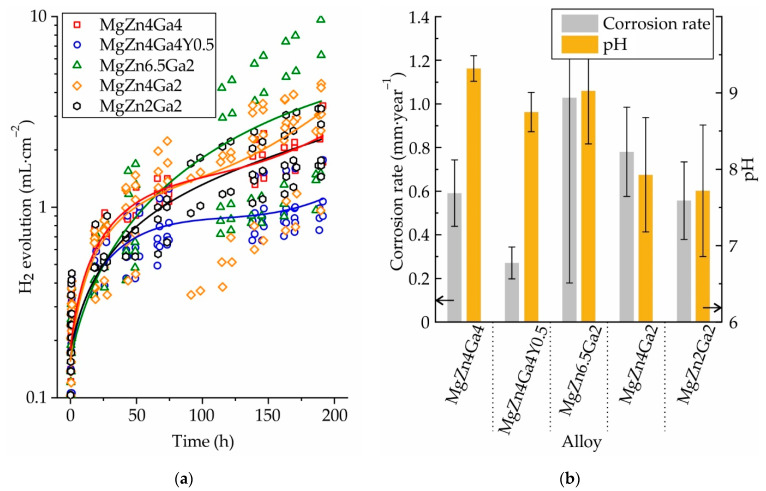
(**a**) H_2_ evolution of the T4 heat-treated Mg–Zn–Ga alloys immersed in Hanks’ solution at 37 °C for 192 h. (**b**) Calculated CRs of the alloys and pH variation of the medium during the immersion corrosion testing.

**Figure 12 materials-14-07847-f012:**
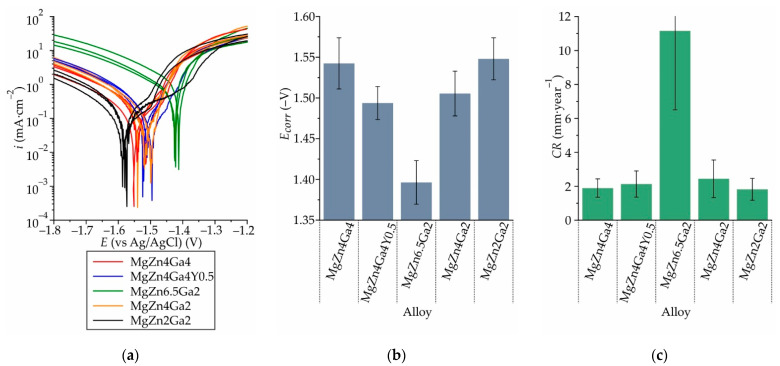
(**a**) Polarization curves, (**b**) corrosion potential (*E_corr_*), and (**c**) corrosion rate (CR) obtained via electrochemical corrosion testing of the T4 heat-treated Mg–Zn–Ga alloys in Hanks’ solution at 37 °C.

**Table 1 materials-14-07847-t001:** Chemical compositions of the prepared alloys.

Alloy	Element Content (wt.%)			
	Mg	Zn	Ga	Y
MgZn4Ga4	Bal.	4.2	4.1	-
MgZn4Ga4Y0.5	Bal.	4.2	4.1	0.4
MgZn6.5Ga2	Bal.	6.5	2.0	-
MgZn4Ga2	Bal.	4.2	2.2	-
MgZn2Ga2	Bal.	2.3	2.3	-

**Table 2 materials-14-07847-t002:** EDS analysis results of the as-cast high-alloyed Mg−Zn−Ga alloys.

Alloy	Phase	Element Content (at.%)			
		Mg	Zn	Ga	Y
MgZn4Ga4	Mg_7_Zn_3_	Bal.	14.9 ± 1.9	9.3 ± 2.1	-
	Mg_5_Ga_2_	Bal.	5.7 ± 0.7	20.3 ± 0.7	-
MgZn4Ga4Y0.5	Mg_7_Zn_3_	Bal.	16.7 ± 2.8	10.1 ± 2.7	-
	Mg_5_Ga_2_	Bal.	7.2 ± 1.1	19.5 ± 1.9	-
	GaY	Bal.	14.3 ± 2.2	31.5 ± 2.4	25.6 ± 3.5
MgZn6.5Ga2	Mg_7_Zn_3_	Bal.	19.8 ± 0.9	4.2 ± 0.6	-

**Table 3 materials-14-07847-t003:** Mechanical and corrosion properties of Mg alloys with Ga contents.

Alloy	Condition	Ref.	Property			
			YS (MPa)	UTS (MPa)	El (%)	CR * (mm/year)
MgZn4Ga4	18 h SSHT at 350 °C	This work	92	211	8.2	0.59
MgZn4Ga4Y0.5	18 h SSHT at 350 °C	This work	97	207	7.5	0.27
MgZn6.5Ga2	18 h SSHT at 350 °C	This work	82	242	10.2	1.03
MgZn4Ga2	18 h SSHT at 350 °C	This work	66	228	12.5	0.78
MgZn2Ga2	18 h SSHT at 350 °C	This work	68	218	15.2	0.56
MgGa4	As-cast	[19]	66	188	7.2	0.24
MgGa5.5	12 h SSHT at 375 °C+ 0.5 h aging 225 °C	[17]	107	181	8.2	-
MgGa5.5	12 h SSHT at 375 °C+ 128 h aging 225 °C	[17]	151	197	4.7	-
MgGa	As-cast	[18]	28	90	3.8	-
MgGa3.5	As-cast	[18]	61	150	5.6	-
MgGa5	As-cast	[18]	61	187	8.4	-
MgGa3.5	Extruded at 300 °C (ratio 10:1)	[18]	158	245	13.7	-
MgGa0.375	As-cast	[22]	-	-	-	0.67
MgGa0.75	As-cast	[22]	-	-	-	0.72
MgGa1.125	As-cast	[22]	-	-	-	0.87
MgGa1.5	As-cast	[22]	-	-	-	1.02
MgGa0.375	12 h SSHT at 350 °C+ 16 h aging 225 °C	[22]	-	-	-	0.60
MgGa0.75	12 h SSHT at 350 °C+ 16 h aging 225 °C	[22]	-	-	-	0.55
MgZn4Ga4	ECAP, 3 passes at 310 °C (CB route)	[5]	168	298	22.8	0.16
MgZn4Ga4Ca0.2	ECAP, 3 passes at 310 °C (CB route)	[5]	165	255	17.5	0.37
MgZn4Ga4Y0.3	ECAP, 3 passes at 310 °C (CB route)	[5]	144	283	18.0	0.22
MgZn4Ga4Nd0.3	ECAP, 3 passes at 310 °C (CB route)	[5]	139	283	27.0	0.30

* The media for CR measurements are simulated body fluid (SBF) [5], 9 g/L NaCl water solution [19], and Hanks’ solution (this work, [5]).

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
