# Peer review of "Effect of Heat Treatment on the Mechanical and Corrosion Properties of Mg–Zn–Ga Biodegradable Mg Alloys"

_materials, 2021, doi:10.3390/ma14247847_

Round 1

Reviewer 1 Report

In this study, 18the influence of different heat treatment regimes on the microstructure and mechanical and corrosion properties of biodegradable Mg–Zn–Ga alloys were investigated. This is an interesting job. This work has a sufficient amount of data, but it seems to be slightly inadequate in terms of chart processing and data repeatability. The author is requested to improve the article according to the following suggestions.

(1) How to measure the Ga/Y atomic ratio (close to 1/1) in line 147 of P4?
(2)  In line 223 of page P7, why select the 200-9 aging? At this time the conductivity has not reached the peak value. According to the text "Owing to the formation of Zn- and Ga-rich precipitates during the aging, the hardness and electrical conductivity increased." in line 214, the conductivity has increased without reaching the peak aging.
(3) The statistical deviation in Figure 8(c) is too large, and objective facts should be respected. For example, the yield strength and ultimate tensile strength of As-cast and 300-18h.

(4) Some figures should be redrawn for better visual effects.. For example Fig. 4 and Fig. 5.

Author Response

Reviewer 1

In this study, 18the influence of different heat treatment regimes on the microstructure and mechanical and corrosion properties of biodegradable Mg–Zn–Ga alloys were investigated. This is an interesting job. This work has a sufficient amount of data, but it seems to be slightly inadequate in terms of chart processing and data repeatability. The author is requested to improve the article according to the following suggestions.

(1) How to measure the Ga/Y atomic ratio (close to 1/1) in line 147 of P4?

Answer: The atomic ratio was determined using EDS. This data was added to table 2.

(2)  In line 223 of page P7, why select the 200-9 aging? At this time the conductivity has not reached the peak value. According to the text "Owing to the formation of Zn- and Ga-rich precipitates during the aging, the hardness and electrical conductivity increased." in line 214, the conductivity has increased without reaching the peak aging.

Answer: The time at which the alloys reached peak electrical conductivity values and hardness was different for the investigated alloys. We agree that peak electrical conductivity values were not reached after 9 h, and a longer time is needed. The 9 h aging time decision was based on hardness measurements mainly because of the relationship between alloys' hardness and strength. However, a longer aging time is possibly needed, but as shown further, aging leads to decreasing elongation at fracture and was inappropriate for designed alloy.

(3) The statistical deviation in Figure 8(c) is too large, and objective facts should be respected. For example, the yield strength and ultimate tensile strength of As-cast and 300-18h.

Answer: We agree with that comment. Maybe the number of tested samples was not enough. However, we said that UTS was not changed for alloys after HT because of the significant statistical deviation. The conclusion about decreasing the YS is based on the trend observed for both MgZn4Ga4 and MgZn4Ga4Y0.5 alloys.

(4) Some figures should be redrawn for better visual effects.. For example Fig. 4 and Fig. 5.

Answer: We do not exactly understand what is wrong with Fig. 4 and Fig. 5. All presented data are visible and readable.

Reviewer 2 Report

The Mg-Zn-Ga concept system is new and interesting to explore. The paper contains new findings, which  are of interest for the scientific community. Gallium indeed shows many attractive properties including the promise for the treatment of certain diseases. However, this potential has yet to be explored and the possible adverse health effects of gallium compounds have to be documented before these alloys can be advanced to clinical trials or even to preclinical testing.

I would suggest to tone down the expression in the last conclusion (line 448-450 “ suitable for applications in low-loaded bone implants because of its good mechanical properties …and low biocorrosion rate)  until convincing data on biocompatibility and absence of long-term adverse effects from Gallium and its compounds are obtained. Satisfactory mechanical and corrosion performance are necessary prerequisites, but insufficient for medical trials.

The other comments/questions are

How the chemical composition was assessed?  The details of the EDS technique used should be provided and the accuracy of the results shown in Table 1 should be given too.  The concentration of trace elements such as Fe, Ni and Cu would be good to show too.

The authors did not identify the existing phases correctly.  The SEM images and the very superficial EDS analysis are not convincing despite the reasonable analysis of the possible phases based on binary diagrams.  The authors might be right about existing phases. However, no clear evidence has been presented and the role of different phases, their distributions and relative volume fraction has not been disclosed. No discussion on the hardening mechanisms, particle strengthening, solid solution strengthening, etc. has been provided. The same comments applies to the corrosion behaviour: the properties are declared and well described (this is appositive note), but are not rationalised in terms of the microstructure and relevant mechanisms of corrosion degradation.    

The tensile stress-strain data shown in Fig 8 a, b and 10a are disappointingly poor.  All curves show different initial slopes, which is likely when the experiments have been carried out without proper strain measurements. As a result, the estimates of the yield stress are not reliable and the values of the elongation to failure are most likely overestimated.   

The previous work by some of the present authors, ref.5, shows that the deformation processing of similar Mg-Zn-Ga alloys yields remarkably better results on the mechanical properties and corrosion rate. I am therefore a bit confused with the motivation of the present work. Please, strengthen this part in the introduction  

Author Response

Reviewer 2

The Mg-Zn-Ga concept system is new and interesting to explore. The paper contains new findings, which  are of interest for the scientific community. Gallium indeed shows many attractive properties including the promise for the treatment of certain diseases. However, this potential has yet to be explored and the possible adverse health effects of gallium compounds have to be documented before these alloys can be advanced to clinical trials or even to preclinical testing.

(1) I would suggest to tone down the expression in the last conclusion (line 448-450 "suitable for applications in low-loaded bone implants because of its good mechanical properties …and low biocorrosion rate)  until convincing data on biocompatibility and absence of long-term adverse effects from Gallium and its compounds are obtained. Satisfactory mechanical and corrosion performance are necessary prerequisites, but insufficient for medical trials.

Answer: We agree with the reviewer suggestion and change this sentence to “Thus, MgZn4Ga4Y0.5 after 18 h of SSHT at 350 °C was recommended for applications in low-loaded bone implants because of its good mechanical properties (UTS = 207 MPa; YS = 97 MPa; and El = 7.5%) and low biocorrosion rate (0.27 mm/year).

The other comments/questions are

(2) How the chemical composition was assessed?  The details of the EDS technique used should be provided and the accuracy of the results shown in Table 1 should be given too.  The concentration of trace elements such as Fe, Ni and Cu would be good to show too.

Answer: The chemical composition of the alloys was determined using energy-dispersive X-ray spectroscopy (EDS) on metallographic sections with an analysis area of 1 mm2. For each specimen, three areas were analyzed. This information was added to the article text. The accuracy of the method is close to 0.1 wt%, and because of that, the second decimal place was deleted from table 1. In our practice, we compare the conventional optical emission spectrometry with EDS analysis results and achieve a good correlation.

As for the concentration of trace elements, the method's accuracy is not appropriate for its determination. Furthermore, the optical emission spectrometry is not suited for precision determining impurities level for high purity alloys. Therefore, we suggest that the summary content of impurity is not higher than for raw magnesium (<0.05 wt%).

(3) The authors did not identify the existing phases correctly.  The SEM images and the very superficial EDS analysis are not convincing despite the reasonable analysis of the possible phases based on binary diagrams.  The authors might be right about existing phases. However, no clear evidence has been presented and the role of different phases, their distributions and relative volume fraction has not been disclosed. No discussion on the hardening mechanisms, particle strengthening, solid solution strengthening, etc. has been provided. The same comments applies to the corrosion behaviour: the properties are declared and well described (this is appositive note), but are not rationalised in terms of the microstructure and relevant mechanisms of corrosion degradation.    

Answer: (3.1) The EDS analysis was not superficial. For example, we analyze no less than 15 points for each phase and calculate the mean composition. Nevertheless, we agree that XRD data can be helpful, and in our further investigation, we use it. We have the XRD patterns for Mg-8 wt.%Zn – 8 wt.%Ga and Mg-15 wt.%Zn – 8 wt.%Ga alloys that approve our suggestion about investigated alloys phase composition.

Answer: (3.2) Maybe a discussion of the hardening mechanisms, particle strengthening, and solid solution strengthening is not in-depth. However, the hardening effect of Zn and Zr and were discussed. As for the influence of precipitates on the hardening, the TEM analysis is needed, and this can be the goal of our further investigations. The influence of the phases on corrosion behavior is also discussed. Nevertheless, we add some points to the discussion part of the article in order to improve this section.

(4) The tensile stress-strain data shown in Fig 8 a, b and 10a are disappointingly poor.  All curves show different initial slopes, which is likely when the experiments have been carried out without proper strain measurements. As a result, the estimates of the yield stress are not reliable and the values of the elongation to failure are most likely overestimated.   

Answer: The investigated alloys have different morphology of the second phase (in some samples, the second phase is not present). This factor affects the mechanical properties, including the elastic modulus. We see this in the stress-strain curves. Comparison of the investigated alloys with the alloys after hot extrusion makes no sense because after casting and (or) heat treatment, the second phase has a different morphology and distribution. However, almost the same elastic modulus after hot extrusion is associated with the small size and uniform distribution of the second phase precipitates.

(5) The previous work by some of the present authors, ref.5, shows that the deformation processing of similar Mg-Zn-Ga alloys yields remarkably better results on the mechanical properties and corrosion rate. I am therefore a bit confused with the motivation of the present work. Please, strengthen this part in the introduction  

Answer: The goal of the work was rewritten to “Mentioned earlier advantageous features of Mg−Zn−Ga alloys make it suitable for bone implant applications for osteosynthesis. Furthermore, heat treatment can promote the decrease of alloys corrosion rate compared to alloys after deformation processing. Thus, this study aimed to investigate the influence of heat treatment on the microstructure as well as mechanical and corrosion properties of Mg–Zn–Ga alloys.

Round 2

Reviewer 1 Report

All the pictures in the text, the curve and the text overlap, this is a problem that must be modified.

Author Response

Figures 4 and 5 are corrected in accordance with the reviewer's suggestions.